# Patient death and nurses' coping strategies: Perception of nurses at a tertiary referral hospital in Kenya

**Peris Wategi Kiarie**[1]*, **Gabriel Okombo**[1], **Wambui Makobu**[2], **Joel Ambikile**[3], **Mary B. Adam**[2,4,5]

1 College of Health Sciences, AIC Kijabe Hospital- Kiambu, Kenya, 2 Community Health, Maternal and NewBorn, AIC Kijabe Hospital- Kiambu, Kenya, 3 School of Nursing, Muhimbili University of Health and Allied Sciences, Tanzania, 4 Department of Research, AIC Kijabe Hospital- Kiambu, Kenya, 5 The African Consortium for Quality Improvement Research in Frontline Healthcare (ACQUIRE), Nairobi, Kenya

* wategiperis46@gmail.com

## Abstract

In healthcare facilities, patient deaths are a common occurrence, exposing nurses to diverse behavioral and emotional reactions, particularly within the context of resource constraints in Kenyan healthcare settings. This study aimed to investigate the experiences of nurses at AIC Kijabe Hospital in Kenya regarding patient death and their coping strategies. The focus group discussions sought to understand factors influencing nurses' reactions to death, assessing the adequacy of their basic training in preparing them for coping, exploring the determinants of their coping strategy choices, and gathering recommendations for enhancing coping mechanisms. Employing qualitative research, six focus group discussions were conducted with 50 nurses from various hospital departments, including the emergency department, medical and surgical wards, intensive care unit, and maternity ward. After data collection, the information was transcribed verbatim and analyzed using a thematic analysis approach with inductive coding. Two researchers independently coded the data. A code was an identifying term for a specific part with emphasis on the aspect being investigated. Subsequently, the research team met to compare the codes and reached a consensus on the best interpretation of the data codes. The coding was then categorized into themes and Subthemes. The study findings revealed four overarching themes: individual process, institutional process, work team relationships, and educational gaps. In the individual process, nurses disclosed factors influencing their reactions to and feelings about death, encompassing Stress injuries, views of life (pessimism or optimism), cultural background, religious beliefs, and self-drive/self-management. Nurses expressed immediate emotional responses to the word "death" and conveyed the emotional toll of losing patients. The institutional process delved into how the hospital assisted nurses in coping with patient deaths, revealing a lack of support or guidance in selecting coping mechanisms. Nurses advocated for psychological

**Data availability statement:** The data underlying this study contain sensitive, confidential information from study participants. According to the ethics approval granted by Kijabe Hospital's Institutional Scientific and Ethical Review Committee and the participants' consent, the full data cannot be made publicly available. However, data will be made available upon request to qualified researchers for the purpose of replicating the results. Requests should be sent to the Kijabe Hospital's Institutional Scientific and Ethical Review Committee at researchcoord@KijabeHospital.org. A data-sharing agreement will be required.

**Funding:** Kijabe Hospital is a Mission hospital with a focus on underserved populations. An internal small grant of 500 USD was given to cover ethics submissions, and no author received any salary support to do this work. The funders had no role in study design, data collection and analysis, decision to publish, or preparation of the manuscript.

**Competing interests:** The authors have declared that no competing interests exists.

support, support groups, and counseling sessions. Work relationships and educational gaps were also featured, with nurses emphasizing the deficiency in training on the emotional and psychological aspects of coping with death. They advocated for enhancements in educational preparation to better equip nurses for the emotional challenges intrinsic to their profession.

## Introduction

Death is an unavoidable reality in healthcare, especially in environments where critically ill patients are treated. The experience of patient death affects not only the dying individual and their relatives but also the healthcare providers, particularly nurses, are profoundly affected [1]. Nurses who provide continuous care to dying patients are particularly vulnerable to emotional and psychological stressors. These can range from feelings of compassion, sadness, helplessness, to anxiety and in some cases, severe emotional distress [2]. The emotional burden is often more intense for nurses, due to their close proximity to patients and the frequency of patient deaths. However, while nurses are frequently affected by these experiences, other healthcare professionals, including physicians, are also impacted, though the nature of their involvement may vary [3]. The emotional toll of caring for dying patients often leads to psychological distress, which in turn affects both their mental health and the quality of care they provide to other patients [2,4,5]. As a result, nurses often adopt coping mechanisms that involve empathy, sharing emotions with patients and their families, and using both personal and professional resources to manage their feelings [4]. Yet, negative emotional reactions if left unaddressed can impair the quality of care, especially for terminally ill patients. Research shows that creating a supportive work environment, with resources tailored to nurses' needs, is essential for promoting effective coping strategies [6]. Studies also emphasize the importance of evaluating and enhancing training programs designed to prepare nurses for emotionally challenging situations, particularly death-related distress [7,8]. Effective training equips nurses with skills that help prevent burnout and address psychiatric symptoms, improving their ability to manage the emotional realities of death in healthcare settings [4,7,8].

Several factors shape nurses' responses to patient death including personal attributes such as age, years of experience, communication skills, and emotional resilience. Additionally, the circumstances surrounding the patient's death, including whether it is expected or occurs suddenly, play a significant role in the emotional response [9–11]. Younger, less experiences nurses often struggle more with death anxiety, while more experienced nurses display better emotional regulation and resilience [7,9]. Cultural values, beliefs, and expectations around death further influence how nurses cope with and process patient death. For instance, nurses often face higher emotional distress when caring for young patients or when patient death occurs unexpectedly [11].

Training plays a critical role in shaping nurses' attitudes and emotional preparedness for dealing with death. Studies reveal that many nurses feel inadequately

prepared by their basic training to handle the emotional complexities of death and dying [12,13]. Although end-of-life care is an integral part of nursing education, some programs fail to provide sufficient hands-on experience or communication skills necessary to support dying patients and their families effectively. Training programs that incorporate both theoretical knowledge and experiential learning in end-of-life care have been shown to improve nurses' ability to manage the emotional and practical demands of death in clinical settings [14–19]. Such training also fosters the development of positive coping mechanisms and emotional resilience in nurses as they transition from novices to experienced professionals [16,17].

Nurses utilize various coping strategies to manage the emotional stress associated with patient death, including seeking social and professional support, engaging in hobbies or physical activity, and spiritual or religious practices [20,21]. Some nurses practice distancing, a strategy that involves maintaining emotional detachment from dying patients, while others find comfort in talking to colleagues or counselors, reflecting on their experiences, or turning to faith-based or religious support systems [2,11,16,22]. Despite these coping strategies, institutional support is essential for addressing nurses' emotional burdens to enable them to continue providing high-quality patient care [10]. Approaches like the Stress First Aid (SFA) model, originally developed for use in high stress professions such as the military, have been adapted for healthcare settings. This model provides a framework for identifying and managing stress injuries among nurses. It classifies stress reactions into four distinct types: traumatic injury, loss, moral injury, and wear and tear – each of which can be applicable to nurses experiencing patient death [23]. Integrating SFA into healthcare systems can provide frontline workers with a language and structure to recognize early signs of emotional distress and seek or offer timely support [24].

Although there is ample research on death coping strategies among nurses in high income countries, these studies often take place in well-resourced healthcare settings where nurses have access to necessary emotional and institutional support. In contrast, nurses working in low- and middle-income countries (LMICs) like Kenya, often operate under resource -constrained environments, with limited access to psychological support, continuing professional development, or structured debriefing sessions following patient deaths [25]. Moreover, Kenya presents a unique context due to its cultural diversity and varying beliefs surrounding death [26], which significantly shape how nurses experience, interpret and cope with patient death. Despite this complexity, literature exploring death anxiety and coping strategies within Kenyan healthcare settings is limited. Most existing studies from Sub-Saharan Africa fail to capture the nuanced ways in which cultural differences, institutional and personal factors intersect to influence how nurses respond and cope with death. This gap suggests that the emotional burden faced by nurses in Kenya remains underexplored and inadequately addressed in policy and practice.

This study therefore aimed to fill this gap by exploring patient death and coping strategies among nurses working at AIC Kijabe Hospital – a tertiary referral hospital in Kenya. Specifically, it sought to identify factors influencing reactions to patient death among Kenyan nurses, assess their perceptions of how well their basic training prepared them to cope with patient death, investigate the coping strategies they chose, and gather their recommendations on how to better support nurses in dealing with patient death. By situating the research within a LMIC hospital, this study offers practical insights that can inform nurse training, institutional policy, and the development of culturally sensitive support systems, thereby contributing to the improvement of the quality of care provided to dying patients

## Materials and methods

### Setting

The study was conducted at AIC Kijabe Hospital, a Christian, faith-based institution whose mission is grounded in providing compassionate and holistic care. It is a level 5B tertiary teaching and referral facility sponsored by the Africa Inland Church (AIC) Kenya. The hospital is located in rural Kenya, in Kiambu County, approximately an hour's drive from Nairobi, the capital city, towards Nakuru. AIC Kijabe Hospital provides both inpatient and outpatient services with a bed capacity of 350. The hospital performs over 8,000 surgical procedures annually and serves more than 130,000 outpatients from

Kenya and beyond. Inpatient services include intensive care, medical, surgical, oncology, obstetrics and gynecology, palliative care, pediatrics, and neonatal intensive care units. The hospital employs approximately 1,000 staff, including nurses, who constitute the largest healthcare workforce.

## Design, sample and data collection

The study employed a qualitative phenomenological design, chosen to explore participants' lived experiences and perceptions in depth. This design was appropriate given the emotionally charged and personal nature of coping with patient death, which requires capturing the subjective meaning and essence of nurses" experiences. The study sample included qualified nurses actively employed at AIC Kijabe Hospital, specifically those providing care in various critical units such as the emergency department, medical and surgical wards, and different intensive care units (Adult ICU & HDU, Pediatric ICU and HDU, Maternity ICU and HDU). Participants were recruited based on specific criteria, including having at least six months of employment since their initial certification, actively providing direct patient care, and having cared for one or more dying patients in the previous six months. Additionally, participants needed to be available to be released from their shift for study involvement. Nurses with prior training in palliative care were intentionally excluded to focus on nurses who may not have specialized palliative care knowledge. This decision was based assumption that nurses working in critical care settings, where they frequently encounter critically ill patients, would have developed informal coping mechanisms through their general nursing experience, rather than structured palliative care instruction. The aim was to explore the emotional and psychological impact on nurses who regularly face similar end of life care challenges but without the benefit of formal preparation. These inclusion and exclusion criteria ensured that participants were both relevant to the study context and able to provide nuanced insights into their experiences.

A purposeful sampling method was used to select qualified nurses who met the inclusion criteria and were willing to participate in focus group discussions (FGDs). Individual interviews were not conducted; FGDs served as the sole data collection method. This approach was selected because FGDs promote dynamic interaction among participants, enabling them to reflect and articulate their personal experiences while also building on the experiences of others. Nurse leaders in each department/ward/unit facilitated the recruitment process by contacting potential participants through departmental WhatsApp groups and delivering formal invitation letters. While the initial plan was to conduct FGDs with 6–8 participants, in two instances, 14 eligible and willing participants attended the session. Given the relevance of their experiences and their voluntary presence, none were turned away. Instead, the sessions were carefully moderated to ensure that all participants were given an equal opportunity to share their perspectives, and the duration of the discussion was adjusted accordingly. Each FGD lasted between 1 and 2 hours, depending on the group size and level of engagement. Although traditional qualitative guidelines recommend smaller group sizes for FGDs (typically 6–8 participants), larger groups of up to 12 or more are permissible when managed appropriately, especially for topics that benefit from a wider range of viewpoints [27]. The research team ensured that active facilitation techniques were employed to manage the larger group effectively and maintained the quality of data collection. In total, six FGDs were conducted between March and May 2023. Data collection continued until saturation was achieved, defined as the point at which no new themes, insights, or patterns were emerging. This was recognized during the fifth and sixth FGDs, where responses became increasingly repetitive and no additional meaningful data were identified. The study obtained ethical approval from the Institutional Review Board (IRB) of AIC Kijabe Hospital and the National Commission for Science, Technology and Innovation (NACOSTI). All Participants provided voluntary written consent prior to the discussions.

FGDs were conducted in a quiet, neutral venue within the hospital premises. Participants were offered breakfast before the sessions, and transport home was facilitated. Each discussion was co-facilitated by two members of research team – one serving as the moderator, using a predetermined focus group discussion guide "Guide in S1", and the other acting as observer and note taker. All discussions were audio recorded with participants' consent, and subsequently transcribed verbatim to ensure accuracy.

Although the researchers were staff members within the same institution, they did not hold supervisory roles over the participants and were not assigned to the same departments. This minimized potential bias or pressure to respond in a particular way. The third author moderated all FGDs, while the first and second authors alternated roles as observers and note-takers across sessions.

## Data analysis

Following transcription, the first and second researchers independently reviewed the transcripts from the first two FGDs and inductively generated meaningful fragments, an approach that is well suited for small scale, health related qualitative research [28]. These fragments were then coded and analyzed thematically, with collaborative comparison by all the researchers to ensure alignment and credibility. The initial codes were organized into categories and further developed into themes through iterative process involving constant comparision across the transcripts to refine and identify recurring patterns, differences and emerging concepts. This thematic approach analysis strengthened the trustworthiness and richness of the findings.

To enhance validation, the preliminary findings were presented to nurses working in the facility during their regular meeting. Nurses confirmed that the themes reflected their experiences and strongly supported the recommendations. Excel software was utilized for managing and organizing the coded data during the analysis process.

## Findings

**Participants Characteristics.** Table 1 summarizes the characteristics of the 50 study participants, the majority of whom were female (78%), held diplomas (90%), and had more than three years of work experience (just over 50%).

## Main themes

The study identified four overarching themes: individual process, institutional process, work team relationships, and educational gaps. Each of these themes further encompasses sub-themes which provided a comprehensive understanding of the various aspects related to the study's purpose.

**Table 1. Characteristics of participants.**

|  |  | Participants n (%), N=50 |
| --- | --- | --- |
| Gender | Male | 11 (22) |
|  | Female | 39(78) |
| Age | 20- 25 years | 8 (16) |
|  | 26-30 years | 30 (60) |
|  | Above 30 years | 12 (24) |
| Departments represented | Adult Intensive care unit (ICU) | 8 (16) |
|  | Maternity | 2 (3) |
|  | pediatric ward | 11 (22) |
|  | pediatric ICU | 11 (22) |
|  | Medical/ Surgical Ward | 11 (22) |
|  | Casualty | 7 (14) |
| Level of Training | Diploma in nursing | 45 (90) |
|  | Advanced diploma in nursing | 5 (10) |
| Level of experience | Above 3 years | 26 (52) |
|  | 2-3 years | 19 (29) |
|  | Below 2 years | 5 (10) |

**Theme one: Individual process**

**Sub theme 1: Stress Injuries**

Nurses described a range of immediate emotional responses associated with death, using terms such as "loss," "grief," "pain", "end of life," "emotional suffering," and "separation." Some characterized death as "cruel" and "the worst animosity." These responses reflect the psychological impact and cumulative emotional toll of caring for dying patients.

Nurses acknowledged that some deaths impacted them more deeply than others. They stated that the circumstances surrounding a death, such as the suddenness or expectedness, personal associations, and the cause of death, can influence the emotional response.

*"I was taking care of this child who looked just like my daughter — she was the same age as my daughter. But that child died. I really worked hard to save her, but she still died, and that really traumatized me. It made me wonder: do these deaths bring back old scars or reopen pains that we thought had healed?"*

*"The sad part of working in the critical care unit is witnessing numerous deaths, some of which have left deep scars in our hearts. It is especially painful when young people, simply going about their daily routines, are suddenly taken away by an accident or unexpected event."*

*Many* nurses expressed a deep sense of helplessness and frustration when outcomes could not be changed despite their best efforts. They emphasized the emotional strain of perceived expectations from the relatives that healthcare workers should be able to prevent patient's death.

*"For me, it is the lack of control, because most of the time when you visit a hospital, you expect that the people there know what they are doing and that they will be able to help you with your situation. But when it is now on your side — you do everything, you can, yet you are still unable to control the patient's outcome — and when it ends in death, you are left feeling completely out of control."*

*"If I can talk of disappointment, when you take care of a patient and it happens that patient passes on, you really feel disappointed because maybe you have invested all your knowledge, all your energy to that patient. When you are hit by death, you are left with so many questions_ 'how will I approach the parents?' (If it's a baby), 'What will be the next thing?' You just feel disappointed. At a point when you are at work, you just feel like you just want to go home."*

Nurses working in maternity department expressed particular distress due to the unexpected nature of maternal deaths. They highlighted the psychological contrast between anticipated positive outcomes and the reality of loss.

*"At times you feel shocked- like in cases at maternity, you only expect a client to come in, get their baby, and go home. And then maybe it is like an emergency, then something happens, and the patient dies. You are shocked because this was not the expected outcome."*

Despite some societal assumptions about death being more acceptable in old age, nurses emphasized that all deaths regardless of age carry emotional weight. They acknowledged that repeated exposure to such losses accumulates over time, leaving deep emotional scars that are often hidden.

*"Nurses are suffering, you see us walk and you are like, 'These people are okay.' We are not. The whole smile is here, but deep down, we just wish we could have someone to talk to. Someone to listen to us. Nurses need someone to listen to them."*

Nurses also spoke about internalizing blame after the death of their patients, questioning their clinical decisions and whether they did enough. These self-reflections suggest experiences of moral distress and trauma related stress injuries.

*"Most of the time, you blame yourself and wonder what you could have done differently to save the patient's life."*

These narratives demonstrate the complex emotional burden nurses bear – ranging from grief and shock to helplessness and moral conflict consistent with the range of stress injuries identified in the stress first aid model.

### Sub theme 2: View of life: Optimism and Pessimism

Nurses expressed a personal fear of death, emphasizing the uncertainty of waking up the next day. They indicated that the fear of death serves as a reminder to appreciate life more fully.

*"I feel like death gives a wakeup call and makes you appreciate life more. I don't think people would really appreciate life more if people didn't die."*

Nurses expressed that the unexpected nature of the patient's deterioration contributed to the emotional burden caused by the unforeseen negative outcomes, as their main goal is to see patient recover and go back home.

*"For me, like I had said earlier, the devastation was due to the unexpected death, because this was a patient who came walking."*

*"It is usually the unexpected because your goal is not for this patient to die. You provide care for this patient to feel better and go home. So, what usually affects health workers emotionally is the unexpected death."*

Nurses expressed their belief in miracles and a higher power and indicated that they prayed for their patients.

*"Then I was like, 'miracle works,' so I just kept quiet. She told me that she's going home, and I told her that God is in control of everything."*

Nurses expressed the belief that life itself was a gift and a blessing, and the occurrence of death was part of a larger divine plan.

*"For I believe that every gift given through birth is a blessing. So, when it happens, so you see it as a relief to the family and within God's plan."*

### Sub theme 3: Culture

Nurses mentioned of rituals and practices related to death, such as the viewing of the deceased and overnight vigils, which highlighted the influence of cultural background on individuals' experiences and perceptions of death.

*"You live with other people from different cultures. Some have those rituals whereby the body has to come to the homestead and stay at home for at least a couple of hours or overnight."*

*"Me, where I come from, death is feared, so it is never a good thing."*

### Sub theme 4: Religious beliefs

In the context of the study setting, religious beliefs and spirituality were found to play a central role in shaping nurses' understanding and approach to death and dying. Nurses described death not merely as the end of life in the physical world, but as a transition into the spiritual realm.

*"I can explain death as the process of the end of life in a physical world, and a transition of life into a spiritual world, whereby you cannot continue with your duties the way you were performing them, when you were alive."*

Faith was also a vital coping mechanism for nurses dealing with the emotional burden of patient loss. Attributing the timing and occurrence of death to God's will help nurses accept patient deaths and continue with their responsibilities without becoming emotionally overwhelmed.

*"You comfort yourself that maybe it was God's will."*

*"Maybe it was God's will, because if you take too much upon yourself, you get drained. So you just have to assume it was God's will. It was their time; it was God's purpose."*

### Sub theme 5: Self-management

Coping mechanisms, such as crying, disconnecting quickly, mourning with relatives, self-comfort, listening to music, sharing with friends and prayers were mentioned as ways to deal with the sadness and emotional impact of death.

*"it is hard, but you comfort yourself and tell God...it is okay, you just pray and let it go, by sharing with your colleagues."*

*"For me, I am very emotional. I usually find myself crying-especially when you find yourself working in labor ward, and maybe you have two mothers and their babies are dead. That thing is really bad! For me I can't cope. I tend to get very emotional; I normally go to the in-charge's office to cry."*

*"For me, it is listening to music, going to church to pray, and sharing with friends- having someone to listen to you and letting you explain your feelings to them - that makes you cope faster."*

Nurses also indicated that there are instances where they avoid the relatives after the death of a patient as a way of dealing with the emotional impact

*"For me, I tend to avoid relatives as much as I can after the scenario."*

There were instances where nurses experience a sense of denial regarding the death of a patient, particularly when emotionally invested in the patient's care. As a coping mechanism, some nurses projected this denial by placing blame on the doctors (Table 2).

*''For me, a patient that I was emotionally invested in -after she passed, there was denial and there was projection to the doctors. I felt that it was actually their fault and believed that they could have done more for that patient. So, my denial was projected to the doctors, putting the blame on them."*

### Theme Two: Institutional process

### Sub theme 1: Limited resources

Nurses pointed out the emotional distress they go through whenever they witness a patient die due to unavailability of necessary medical resources and some are left with an enduring mark on their memory.

*"It was very painful when you a see patient die because of lack of a machine, or because of lack of medication… It was an experience that I have never gotten out of my mind."*

**PLOS One**

**Table 2. A table illustrating the meaning of the Individual process theme and its subthemes.**

| Theme | Meaning of the theme | Subthemes | Meaning of the subthemes |
|---|---|---|---|
| **Individual process** | The theme provides a comprehensive understanding of the factors that shape nurses' personal reactions, feelings and coping strategies about patient death. | **Stress Injuries** | The subtheme explains the emotional and mental distress experienced by nurses in response to the challenges and emotional burdens associated with death and the care of dying patients. |
| | | **Optimism and Pessimism** | The subtheme gives an explanation of how nurses viewed life and death which was an interplay between pessimism and optimism. |
| | | **Religious belief** | The subtheme explains how nurses incorporate a spiritual dimension into their understanding of death. |
| | | **Culture** | The subtheme brings an understanding about the impact of nurses' cultural background on their experiences and perceptions of death. |
| | | **Self-management** | The subtheme explains the individualized and self-directed approaches nurses adopt to navigate the emotional complexities associated with patient deaths. |

Nurses implied a sense of helplessness and a lack of available options for advanced medical interventions, highlighting the disparities in healthcare resources compared to developed countries.

*"If the patient was in developed country. maybe he would have benefited from a transplant. But here in Kenya, there is nothing much that could have been done."*

**Sub theme 2: Policies**

Nurses Pointed out the need for organizational policies or systems that support and formalize the rotation of nurses between different hospital departments, with the aim of promoting their emotional well-being and preventing burnout.

*'The in charge in my department has established this rotation so that nurses can get to work in HDU, where the patient are a-bit more stable, and you work there for some time. This change of environment - going to different rotations- gives us a break to recollect ourselves, so that before the next rotation in the department, you are back on your feet."*

*"In my opinion, I think the hospital has a mandate to swap nurses between different department. Like you have been in the ICU, you have seen patients die and die- to a point you feel like you are disconnecting from being human."*

*"A swop, like from inpatient to the outpatient, would really help- to see heathy, walking patient around and to connect and hear from the patient. You know, we don't even take history ourselves; you receive history from relatives and you don't know whether it is correct or not. You never get to hear this patient talk, because by the time they come to ICU, they have escalated. If they can be swapping us around for some duration, that can really help us to recover. Because as much as you have done the specialization, you still feel lonely in those cubicles- like you are in that sad cubicle from the first day of the year to the last day of the year… which makes it hard to connect with the real world. Some of us have been converted to introverts unknowingly …. but you are in a healthcare setting or hospital … this is a profession you are in... where patients do not talk."*

*"I think that policies can be reviewed… Some patients who are admitted in ICU have poor prognosis. Relatives should be well prepared about their prognosis before the patient is escalated to ICU. Sometimes the patient goes into arrest before they are even connected to a machine… so you start resuscitation……which becomes unsuccessful -only to be told that the patient had stage IV cancer…and you feel… you could have been told early."*

Nurses stressed the importance of breaks for nurses to assess their emotional well-being. They requested understanding from the hospital regarding the intensity of situations and the need for nurses to manage their workload effectively and promote job satisfaction.

> "*I think some deaths are more painful to a point that they warrant maybe a day off or maybe a day to rest.*"

> "*Give us a break. A break of understanding. We have injuries in the ward… and those closest to us can assess: how intense is the situation or injury, for that matter? Can she manage the day? Does she need just a few hours, two minutes, or does she actually need a day?*"

> "*You have nursed a baby for more than a month, and according to you, the baby was progressing well. Then the baby's condition changes, deteriorates and dies. They need to give a break to relax your mind and then come back when you are fresh.*"

Nurses expressed a desire for the opportunity to choose their work placements based on personal preferences and skills, emphasizing that such choices could positively impact their mental well-being.

> "*And also, giving nurses a choice about where they serve best could affect how well they manage their duties. Some nurses may be placed in areas like pediatrics, but they feel forced to be there and lack the capacity or passion for that field. While they are trained nurses, their hearts may be elsewhere—like in maternity care. If there were an opportunity to choose where they could best use their skills, such as selecting two or three preferred areas, it might help those who feel stuck in a position just because they need the job.*"

> "*You see, it's not about nurses deciding entirely where they want to work, but rather having evaluations to assess how they're performing in areas like med-surgical or pediatrics. These evaluations could help determine where someone might excel. With that kind of system, nurses could express their preferences and be monitored accordingly, without the fear of being told there's no place for them. The concern is that if I don't want to work here, I might lose my job, so I end up staying just to get paid and survive. I think we need that kind of flexibility and understanding.*"

Nurses expressed frustration with the blame game during audits, where responsibility is shifted to them, even for issues that were beyond their control. They indicated that the blame was particularly pointed during deaths, even when surgical complications were involved. They gave a suggestion of stopping the blame game as a crucial first step. They stressed the importance of audits being used to identify gaps rather than assigning blame.

> "*I'm not sure how to express this, but there's often a push-and-pull aspect when things go wrong—like when you're the one who administered medication, and then there's a blame game. It really requires teamwork. I think everyone should take responsibility and acknowledge that we did our best, even if it wasn't perfect. Maybe it was part of God's plan. The blame game should be eliminated because it's very common, and I've experienced it frequently when someone blames another for an outcome.*"

Nurses expressed discontentment with the current debriefing process, which they indicated that involved filling out forms immediately after a patient's death, and they suggested a more meaningful debriefing session.

> "*But this is what they have labelled debrief, but ideally it is not a debrief. It's actually a cardiac arrest form, which is being used in place of a proper debrief. What I mean is, the form just asks for details about what was done during CPR—how long it took, what was administered, the patient's diagnosis, the outcome, and the equipment used. That's it. Once the form is filled out, it's over. Everyone just leaves—no prayers, no further discussion, nothing.*"

**Sub theme 3: Psychological support**

In exploring how the institution assisted nurses in their choice for coping strategies and in addressing the challenges they face in dealing with patient deaths. Nurses expressed that the institution does not provide any support or guidance in selecting and in their coping mechanisms, and the question led to laughter among the nurses. However, some nurses indicated that there was debriefing in the past but they were not sure whether the practice continued.

*"I feel the same. Some of us get tortured. Like she said, maybe this morning you encountered unexpected situation and lost a patient. We just concentrate with the relatives and you forget about this nurse who has other patients to take care of. She is not stable. I wish there was someone to take the nurse aside and give some emotional support."*

Nurses expressed a sense of being neglected, mentioning that chaplaincy support concentrated on comforting the relatives and often forgot to take care of healthcare providers.

*"The only time I remember the chaplaincy focusing on us was in 2016. That was the last time the chaplaincy team visited the casualty department, and it was after a tragic accident. There were six bodies in casualty, and it wasn't easy. They came to offer support."*

*"Though they say that the chaplaincy offices are open and that we should always go to them if something is bothering us or if we feel we can't handle it, sometimes we don't see the need to reach out."*

*"I think the hospital focuses more on the relatives and not to the staff. For example, when a patient passes away, the chaplaincy only come to encourage the relatives and not the healthcare teams. I think there should be a session for the healthcare givers for them to get encouragement and hope. If a chaplain meets with the relatives, they should also have a session with the staff before they go to the relatives."*

When nurses were asked on what the hospital could provide to help nurses cope with patient deaths, the following suggestions were floated:

Nurses highlighted the importance of having a psychologist available within the hospital to assist them in coping with emotional challenges. They emphasized the proactive approach of therapists in reaching out to nurses, rather than waiting for nurses to seek help. They also expressed concerns about current difficulties in accessing the existing psychologist and stressed the importance of ensuring availability and accessibility for the nursing staff. Suggestion of the psychologist to visit every department, providing mandatory counseling sessions after patient deaths was made.

*"Maybe once every three or six months, the chaplaincy team could hold debrief sessions in departments where death rates are high. This would be very helpful."*

*"The chaplaincy shouldn't just leave it open for staff to visit their office if they have an issue. Most people are too shy to go and discuss their emotional challenges, so no one ends up going. The chaplaincy could organize forums to engage healthcare workers, which would make it easier for nurses to participate as a team".*

Nurses acknowledge that making time for psychological sessions may be challenging, but for their own benefit, they expressed willingness to create time for them.

*"We will have to make time for our interest. Yes, it will be tricky, we understand that it will be tricky, but for our own benefit, we will have to create time for them. You will tell the in-charge, to give you these 30 minutes or one hour, yes, …… that one must work because you are taking care of yourself as much as you are taking care of your patients."*

                    

*"Our mental health should be a priority so that we don't get depressed."*

Support groups for nurses were proposed, emphasizing the benefits of collective encouragement and problem-solving.

*"Like when a patient dies, if we are in a group, you can stomach it, …. you are able to find a way to encourage one another."*

*"A forum for nurses could be organized once a month or every six months, allowing them to express and share their emotions. This might help nurses download their emotional feelings."*

Nurses suggested the need for counseling sessions, particularly after the death of patients (Table 3).

*"The other thing which I think is also good, is to include the primary nurses when you are counselling the relatives because you yourself also needs to be psychologically prepared."*

*"Sometimes people tend to blame others about what happened…. so, it is good if they go through counselling because death brings you down to an extent that you get phobia of nursing particular patients."*

### Theme Three: Work/ team relationships

### Sub theme 1: Peer to Peer relationship

Coping mechanisms through peer engagement mostly included use of injected/dark humor which was mentioned as a way that nurses resulted to, to deal with the sadness and emotional impact of death.

*"We normally talk about it with colleagues. So, I will be at the nursing station...we will talk and talk about it with other colleagues. So, when I go home, I am somehow okay."*

*"For me talking to colleagues, when we debrief each other…. you say what and how you feel. Like in ICU, we normally do that debriefing for everyone to talk about how they feel about that incident. What has happened and how it has affected you and that helps a lot".*

*"Let me start by saying I'm a very talkative person. We often—though it's not necessarily a good thing—cope by joking around. That's my way of dealing with stress. For example, if you're losing a patient or resuscitating one and the mood*

**Table 3. A table illustrating the meaning of the Institutional process theme and its subthemes.**

| Theme | Meaning of the theme | Sub themes | Meaning of the subthemes |
|---|---|---|---|
| Institutional process | The theme addresses the organizational support, structures, and policies within the hospital to assist nurses in coping with emotional challenges and addressing the difficulties they encounter in dealing with patient deaths | Limited resources | The subtheme explains the nurses' experiences of emotional distress when faced with the unavailability of essential medical resources, leading to a patient's death. |
| | | Policies | The subtheme explains the nurses' perspectives on what the organization can do to support and formalize certain aspects of their work environment with aim of supporting their emotional well-being, prevent burnout, and enhance job satisfaction. |
| | | Psychological support | The subtheme explains the nurses' recognition of the importance of having access to psychological support services within the hospital setting, their preferences for proactive and accessible services, and their willingness to make time for such support despite potential challenges. |

*is heavy, I'll say something light-hearted and we'll laugh about it. It's not in a bad way, but rather a way to help lighten the moment, and I think it really helps me."*

Nurses expressed the value of positive reinforcement and mutual encouragement from their teams when facing challenging situations.

*"I've come to appreciate the power of positive reinforcement—patting my partner on the back and them patting mine, saying, 'You did well.' Focusing on the positive has really helped me overcome challenges."*

**Sub theme 2: Motivation of nurses**

Nurses indicated that, while financial considerations are a factor for some individuals, many nurses are still deeply committed to their profession and driven by a sense of purpose and dedication to patient care.

*"And right now, nursing has become very marketable. People are entering the profession for the money—they're not coming for the calling anymore. The sense of calling is gone, and it needs to be revived, starting from the time they begin their training."*

Nurses shared of instances where the death of their patients had an emotional toll in them and that affected their productivity.

*"For me…. I was not able to do anything after that incident …. I was not given a break…. the day was dark."*

Nurses expressed that they often find their motivation to work influenced by a range of emotional experiences, particularly those related to the challenging aspects of patient care. Such as caring for terminally ill patients. Nurses indicated that they sometimes request not to be assigned to specific beds with terminally ill patients, as witnessing their deaths can be traumatizing.

*"There are days that you wake up and don't feel like going to work…. Sometimes you even find yourselves making calls to request that you are not allocated to certain bed numbers… because you know that they are terminally ill and it's just a matter of time… so sometimes it is a demotivation."*

*Yeah, because yesterday you had a patient pass away, and you left another one in critical condition. So, you wake up and think,* Waaaah! *(taking a deep breath),* How's it going to be today?"

*"Tomorrow, I'll have other patients, and if I hold on to this too much, it will pull me back. Sometimes, it can affect your work for an entire week."*

Nurses also described the emotional impact of observing multiple patients die in quick succession and the subsequent struggle to cope, leading to a temporary refusal to report to duty.

*"After we took the young lady for a CT scan, we came back and found the patient had passed away. Then, after some time, the young lady also died. Watching both patients die was really traumatizing. Suddenly, the HDU/ICU was empty, but by midnight there was a mass admission, and after a while, they all passed away too. After that night, I had to refuse to report to duty for a few days—it was just too traumatic. That experience has never left my mind."*

### Sub theme 3: Work environment

Nurses expressed how their working conditions increased the emotional challenges of coping with patient death. They shared feelings of vulnerability and blame particularly where the mistakes were harshly judged, especially to nurses compared to other healthcare professionals.

*"Why is it that when a nurse is involved—without even making a mistake—people immediately think the worst and say, 'These nurses, these nurses'? It makes us feel threatened at times."*

*"Sometimes, as nurses, we feel threatened because a mistake is something that happens unexpectedly—something you didn't plan for. When someone in another role makes a mistake, nurses often know about it, but no one talks about it. On the other hand, if a nurse makes a mistake, the entire hospital seems to make a big deal out of it. That's why we worry about our licenses being at stake. It feels like every day we come to work, we end the day by thanking God, 'Today, I secured my license.' Then, when I wake up the next day, I pray again, 'Lord, please help me not to be jobless today.' We feel very threatened as nurses."*

Nurses expressed concerns about decisions made by those higher up in the healthcare system without considering the practicalities at the ground level. They called for leaders to visit hospital wards to witness the workload and challenges faced.

*"Some decisions are made at a higher level, and we are expected to follow most of them. I wish those decisions could reflect the ideas and experiences from the ground. It would be beneficial if someone could come to the ground and see how the work is actually done."*

High patient load and mismatched staffing levels left nurses physically and emotionally exhausted, compounding their vulnerability when faced with patient deaths.

*"In medical-surgical nursing, they focus on nurse-patient ratios based on numbers rather than workload. Sometimes, you might have five or seven patients, but the intensity of care can feel like more than twelve."*

*"Sometimes, nurses feel overwhelmed because the patient-to-nurse ratio is 1:12. Caring for twelve patients can be very demanding, especially when you have someone who needs to be fed, turned, or given a bed bath, or when some patients have stomach issues. One patient can be particularly heavy on your workload. Yet, from an outside perspective, someone might look at your team and say, 'You have a few patients; you need to take on several more.'"*

Nurses discussed the impact of workload on their basic self-care, which further limited their capacity to cope effectively with emotionally charged situations.

*"At the end of the day, you often find there's no time for a break, and you haven't had lunch. It's funny how few nurses actually get to go for lunch or even bring their own. You only manage to take a break when you're completely worn out, knowing that if you don't, you'll crash. Most of the time, we just manage to have our tea around 12:00. You realize that even lunch is a rush; you're running around, and when there are many relatives needing attention, you feel pressured to take just thirty minutes. You're expected to stay at the nurses' station because a patient's condition can change at any moment."*

### Sub theme 4: Nurse patient relationship

Nurses highlighted facing challenges in handling relatives when dealing with patient deaths. They mention the difficulty in explaining the occurrence of death, especially when relatives may not understand or react emotionally. They indicated of instances where they are shouted at by relatives making it difficult to navigate the emotional aftermath.

*"It's not easy, especially after losing a patient who was under your care. You have to deal with the relatives, and sometimes it's incredibly difficult to handle their emotions. They often don't understand what really happened, and some may even shout at you after the patient has passed. Carrying that burden is very traumatizing."*

*"Then we had to explain the situation to the relatives of the father, and one of the relatives was so upset that they were about to slap the clinician who was trying to explain."*

Nurses also mentioned of having fear of how others, particularly patients' relatives, perceived them after the death of a patient.

*"Imagine losing a patient while you're still responsible for caring for others. Meanwhile, the relatives begin spreading rumors about the ward, so as you walk by, people say things like, 'You're here to kill us just like you killed someone else.' It's not easy; they are simply scared of you. They even start questioning whether the medication you're administering is the right one, making everything much more difficult."*

Nurses often experience psychological distress when relatives blame them for the death of their patients. They face significant challenges in dealing with families after a patient's death, particularly when relatives struggle to understand that medical situations can yield unpredictable outcomes. This lack of understanding can lead to conflicts and emotional distress for the nurses involved (Table 4).

*"I work with adults, and I can share a case scenario where we had a patient who passed away. The relatives couldn't understand that these things happen. Some believed we had neglected their loved one, and others even became aggressive towards us."*

*"She had battled the illness since childhood and was around 14 years old. After her death, the mother came to me and accused me of being the cause of her child's death."*

### Theme four: Educational gaps

### Sub theme 1: Psychological support

Nurses describes the difficulty in witnessing patients succumb to illnesses, when they were relatively new in the field, especially in their first year of training, which made some of them to contemplate quitting nursing due to the emotional

**Table 4. A table illustrating the meaning of the 'Work/Team Relationships theme and its subthemes.**

| Theme | Meaning of the theme | Sub themes | Meaning of the subthemes |
|---|---|---|---|
| Work/ Team Relation- ship | The theme explains the challenges nurses face in their work environment, interactions with patients' relatives, relationships with peers, and the factors that influence their motivation and emotional well-being in the healthcare setting. | Peer to peer relationship | The subtheme discusses how nurses rely on each other for emotional support and coping mechanisms, particularly in dealing with the emotional impact of patient deaths. |
| | | Motivation of nurses | This subtheme explores how emotional experiences and workplace conditions influence nurses' drive and willingness to work. |
| | | Work Environment | The subtheme highlights the challenges nurses face in their workplace, focusing on issues related to blame culture, decision-making processes, patient ratios, and the impact of workload on their well-being. |
| | | Nurse patient relationship | The subtheme explains the challenges nurses face in managing interactions with patients' relatives, particularly in the emotionally charged context of patient deaths. |

toll. The constant awareness that death can occur at any time in their work environment added to the emotional strain, prompting thoughts about pursuing a different career path.

> *"As a health worker you are trying your level best and seeing that patients succumb to their illnesses was not easy and was very traumatizing. I was very new in the field. I think was in my first year. I thought of quitting nursing and do something else because I could not imagine how life will be. It would be an area that I am working and expect death to occur anytime."*

Nurses suggested improvement on the educational preparation of nurses in coping with the emotional and psychological aspects inherent in their profession. They indicated the need to have mental assessment for students rotating in the clinical placements.

> *"I've come to realize that, despite having all this knowledge. I also need to be evaluated, let me say mental assessment during clinical rotation...we don't really have a defined way on how one can be evaluated and to pick if a student is coping well or needs some help."*

**Sub theme 2: Curriculum gap in experiential learning**

Nurses indicated a deficiency in the training, highlighting that the emotional and psychological aspects of coping with death were omitted.

> *"They taught us about death, dying and last offices, but the psychological part no."*

> *"The aspect of taking care of the dying patient was done, but the emotional aspect of how to take care of myself was never touched."*

Nurses highlighted variation in training on handling death and breaking bad news to patients' families (Table 5).

> *"Because we were never taught on how to break bad news.... For some relatives you never know what might be their reaction. Some relatives, even slap you, they scream. Such a thing happened in my department."*

> *"Students should be educated on how to deliver bad news properly."*

## Discussion of findings

This study provides a profound understanding of the emotional and professional challenges nurses face when coping with patient death. Our analysis reveals that this experience is not a single event but a complex interplay of individual

**Table 5.  A table illustrating the meaning of the educational gaps theme and its subthemes.**

| Theme | Meaning of the theme | Sub themes | Meaning of the subthemes |
|---|---|---|---|
| Educational gap | The theme explains the deficiencies in the training and education provided to nurses, specifically in addressing the emotional and psychological aspects of coping with death. | Psychological support | The subtheme highlights the emotional challenges faced particularly by nurses who are in the early stages of their careers, including nursing students and the need for psychological support mechanisms to address the emotional toll caused by death of patients. |
| | | Curriculum gaps in Experiential learning | The subtheme explains the existing gaps in the educational curriculum for nurses, particularly in the areas of emotional and psychological preparation for dealing with death and dying patients. |

emotional responses, institutional support structures, team dynamics, and foundational educational. "Table 1". The finding demonstrate that patient death constitutes a profound professional and emotional challenge for nurses, one that transcends clinical circumstances and exposes critical gaps in support systems. A critical finding is that the emotional toll on nurses is severe and persistent, whether a death is sudden or expected, and that the current support systems are largely inadequate, relying too heavily on individual resilience rather than structured, systemic support.

The principal finding of our study was the profound impact of what can be conceptualized as "stress injuries." Nurses in our study reported intense emotional reactions, including grief, despair, anger, and helplessness following patient death. This is consistent with a large body of literature documenting the adverse effects of these reactions on both personal well-being and the ability to provide quality care [4,5,29,30]. Our study adds to this body of work by illustrating the depth of trauma nurses experience, particularly when confronted with young patients or those in maternity departments, where positive outcomes are expected. This aligns with a study conducted in a large women's and child healthcare hospital in Singapore, where pediatric nurses similarly reported heightened emotional responses to patient death [16]. This comparison highlights the universality of emotional trauma related to death in pediatric and maternity settings. Additionally, nurses in our study highlighted the emotional toll of losing patients, even the elderly, noting the emotional scars left by witnessing deaths, particularly those of young individuals. Many nurses also reported feelings of self-blame and trauma following the death of their patients, often questioning whether they had done enough or made mistakes in their care. This process of self-reflection and internalized blame further contributes to what has been described in the literature as moral distress- the psychological discomfort that arises when individuals feel constrained in their ability to act according to their ethical standards [23,31]. The SFA model, developed to support individuals in high-stress occupations such as healthcare, offers a useful framework to understand these emotional reactions. It categorizes stress injuries into four types: traumatic injury, loss, moral injury, and wear and tear [24]. Nurses in our study displayed signs of all these domains, from traumatic stress following unexpected patient deaths to moral injury in situations where they felt ethically conflicted to wear and tear from chronic exposure to suffering. Integrating such models into support interventions can help institutions identify varying stress responses and offer tailored support to nurses. These findings emphasize the critical need for structured psychological support systems, particularly those informed by models like SFA, to help nurses process grief and maintain emotional well-being.

These intense emotional responses are not uniform but are filtered through a nurse's individual psychological and cultural lens. We found that nurses' reactions to death were influenced by a dynamic interplay of optimism and pessimism. Some expressed a personal fear of death and emphasized its role in reminding them to appreciate life more fully. In a study conducted through focus group discussions with second- and third-year nursing students, fear emerged as the primary theme they expressed, showing that even early in training, healthcare workers struggle with the concept of mortality [32]. Despite having positive intentions to provide the highest quality care for patients facing death, healthcare professionals often harbor fears related to death that negatively affect their attitudes [4]. Pessimistic views were also apparent, with nurses sometimes perceiving death as a relief, particularly when medical costs become overwhelming. This view reveals a nuanced moral and emotional conflict, as nurses reconcile their duty to preserve life with the economic realities faced by many families. Unexpected patient deterioration added to the emotional burden, as their primary goal is to see patients recover and return home. A literature review by [33] indicated that younger nurses consistently reported stronger fears of death and more negative attitudes towards caring for dying patients. This insight aligns with our findings, where early career nurses were more likely to express emotional vulnerability and pessimistic outlooks. These comparisons suggest that experience, mentorship, and emotional resilience training are crucial in shaping more balanced responses to death.

Furthermore, nurses incorporated a spiritual dimension into their understanding of death, frequently viewing it as the end of life in the physical world and a transition into the spiritual realm, and found comfort in prayer and the belief in a divine plan. In a study conducted with pediatric nurses, similar findings were reported: nurses believed death was part of God's will, and found comfort in knowing that their deceased patient was no longer suffering. These beliefs provided them

with relief and helped them cope and move on [16]. The incorporation of religious belief not only aided personal resilience but may also support culturally competent end-of-life care in settings where faith and spirituality are deeply embedded in daily life and healthcare interactions. Given that this study was conducted in a faith-based institution, where Christian values guide the organizational mission and caregiving approach, it is likely that the institutional culture further reinforced participants' resilience through spiritual coping. This highlights an opportunity for healthcare systems to consider integrating spiritual care training into nursing education and practice, ensuring nurses are supported in both their professional and spiritual roles, particularly in faith-based institutions.

Cultural background also significantly shaped nurses' experiences and perceptions of death. Participants described mourning rituals such as the viewing of the body and overnight vigils, demonstrating the role of cultural norms in providing a shared framework for processing loss. Given the limited research on culture and death anxiety in nursing [4], our study underscores the significance of our findings. By capturing how culture influences emotional responses, this study contributes to a growing recognition of the need for culturally sensitive support systems especially within Kenya's diverse cultural landscape.

In response to these compounding pressures, nurses develop a range of personal coping mechanisms, both adaptive and avoidant. They reported strategies including crying, emotional detachment, avoiding dying patients and, in some instances, their relatives after the death of a patient, mourning with relatives, self-comfort, listening to music, sharing with friends, keeping busy, and engaging in prayer. These findings align with the results of several studies [2,11,13,30,34,35]. A particularly salient finding was the use of peer support and the use of injected/dark or "gallows" humor within the work team to manage grief, a common and recognized coping strategy that provides momentary relief and camaraderie. Nurses also expressed the value of positive reinforcement and mutual encouragement from their teams when facing challenging situations [16,35–37]. However, these individual and peer-level strategies are often insufficient to counter systemic challenges, representing a fundamental contradiction where personal resilience is expected to compensate for institutional shortcomings.

This reliance on personal coping exists within an institutional context that often amplifies, rather than alleviates, stress. A significant source of moral distress identified by our participants was the helplessness caused by limited resources, where patient died due to the unavailability of essential medical resources. These findings align with a study conducted with healthcare professionals in Togo, where participants reported emotional challenges stemming from frustrations due to the lack of appropriate medical equipment and supplies needed to effectively manage patients [30].

Beyond resource constraints, the institutional environment was often perceived as unsupportive. Nurses reported a pervasive "blame culture" where they felt disproportionately blamed when mistakes occurred, compared to other healthcare professionals, and expressed being overwhelmed by high patient ratios, advocating for a shift in focus from numerical targets to actual workload and patient acuity. These factors collectively negatively impacted nurses' motivation and productivity. Some nurses avoided assignments with terminally ill patients due to emotional trauma. Multiple patients' death in quick succession sometimes led nurses to temporarily avoid work. The literature shows that the consequences of a failure to help nurses build strong coping strategies to deal with death anxiety may include nurses leaving their positions, poor communication, decrements in personal health and quality of life, and a reduction in the quality of patient care [4,5,38]. The emotional burden is further compounded by stressful interactions with patients' relatives, who sometimes shouted at or blamed nurses for patients' death, a form of mistreatment previously documented to be higher among nurses and those working in emergency and outpatient departments [39].

The institutional support nurses desperately need is often absent, a gap that begins in their foundational education. Our participants identified a critical curricular gap in experiential learning and psychological support, indicated that their training lacked a focus on the emotional and psychological dimensions of patient death. This left them unprepared for the profound psychological toll, a finding confirmed by numerous studies [8,14,16,17,19,30,32,34,35]. This inadequate preparation is linked to higher death anxiety [7] and, as our study found, leads some nurses to contemplate quitting the

profession, highlighting the urgent need for structured psychological support systems. This supports our participants' calls for improvements in educational preparation, including mental health assessments for students during clinical placements. Our findings extend this literature by emphasizing not only the need for preparation but also the timing and depth of such support. Integrating leadership and organizational support to foster emotional resilience and reduce burnout among healthcare professionals is also important [40]. While several studies have advocated for reinforcing and training coping strategies within nursing education [6,10,19,30,41], our participants suggested that these should be introduced early and integrated longitudinally throughout training. This perspective is reinforced by a literature review of 38 articles, which found that strengthening adaptive and individualized coping strategies and promoting self-care during training would offset burnout [4]. Simulation has also been promoted as a preparatory tool. A review of 16 articles [35] found that the use of simulation in training helped enhance students' confidence and emotional preparedness by improving documentation, communication and technical responses to patient death. However, our study participants echoed concerns raised in the literature that simulations did not fully replicate the emotional realities of encountering patient death in real life, limiting their utility in building true psychological resilience [35]. Importantly, both the current study and prior research recommend more holistic and experiential forms of death education. Reflective activities, peer discussion, art therapy, and imagination – based exercises have been proposed as complementary strategies to help students process their emotions [14,16,35]. Our participants strongly supported such approaches, noting that opportunities for emotional expression and guided reflection were rarely offered but would have been helpful. Committed nursing leadership at the ward level also emerged as a valuable element, with our findings confirming earlier recommendations that such leadership can play a significant role in normalizing discussions about death and fostering a supportive environment [41]. Finally, both the current study and the reviewed literature [4,12,13,33] emphasize the value of expanding death education, with implementation not only during initial training but also in orientation for new nurses and as part of ongoing professional development. Such sustained exposure could reduce death anxiety, improve attitudes towards caring for dying patients, and enhance emotional resilience in both novice and experienced nurses.

In light of these findings, our participants proposed concrete institutional solutions that align with prior research. Their recommendations for policies and psychological support include accessible counseling, support groups, rotation policies to reduce prolonged exposure to dying patients, scheduled breaks and understanding from supervisors, choice of work placement, ending the blame culture and meaningful debriefing sessions. These recommendations are consistent with previous studies that describe the value of chaplaincy services (religious-based support), planful problem solving (which involves strategizing on how to manage situations effectively, such as rotating assignments to lessen the time a given nurse spends with dying patients), momentarily regrouping (taking short breaks to decompress or leaving the workplace altogether), and grief support groups as essential support structures [13,35]. Implementing these changes is not merely an administrative task but a moral imperative to support the workforce.

## Conclusion

Three fundamental truths emerge from our findings: First nurses often form strong emotional connections with patients, and the emotional toll of patient deaths persists regardless of whether the death was sudden or expected. Second, existing coping mechanisms reveal systemic contradictions. While nurses develop personal resilience strategies, from spiritual grounding to peer support, these individual solutions cannot compensate for an institutional lack of support. The very fact that nurses emphasize the need for psychological support, such as accessible sessions with a psychologist and the establishment of support groups for collective encouragement, underscores this gap. Third, nurses highlight the importance of enhanced education on the emotional aspects of handling patient death. The implication of these findings points to a pressing need for systemic changes in both the healthcare workplace and nursing education. Addressing the psychological impact of patient deaths through structured support mechanisms could promote nurses' well-being and potentially enhance retention.

## Study limitations

Despite its value in generating rich data and exploring the meaning of phenomena, the qualitative design of this study is not suited for testing hypotheses or inferring causal relationships. Consequently, the study is limited in its ability to establish cause-and-effect connections. Additionally, the findings may be open to multiple interpretations and may not be broadly generalizable beyond similar settings. Furthermore, the study was conducted in a single hospital, which may limit the transferability of the results to other healthcare organizations with differing cultures, resources, or support structures.

While participants were drawn from diverse departments, providing a broad perspective on emotional experiences across clinical settings, the data were not categorized by department, and the focus group discussions included participants from multiple units. This limited our ability to systematically compare the experiences of nurses across different clinical environments, such as high-intensity units versus general wards. As a result, it was not possible to draw specific conclusions about how the nature of each healthcare setting may influence emotional responses and coping strategies.

Future research that explicitly categorizes participants by department and explores unit-specific dynamics would offer deeper insights into how particular clinical environments shape nurses' psychological experiences and support needs when coping with patient death.

## Recommendations

Based on the findings and implications of this study, the following recommendations are proposed:

1. Healthcare leaders should prioritize the establishment of comprehensive psychological support programs for nurses. These programs should include regular accessible sessions with psychologists, support groups for collective encouragement, and counseling services tailored to address the emotional toll of patient deaths.

2. They should also facilitate the formation of peer support networks among nurses. These networks can provide a platform for sharing experiences, offering mutual support, and fostering a sense of camaraderie among colleagues facing similar challenges.

3. Nurse educators should revise nursing curricula to include enhanced education on the emotional aspects of dealing with death. This could involve incorporating modules or courses focused on coping strategies, self-care techniques, and effective communication skills for managing emotions in challenging situations.

4. They should also work to normalize help-seeking behaviors among nurses. Creating a culture where seeking support for emotional well-being is encouraged and destigmatized can empower nurses to prioritize their mental health and seek assistance when needed.

5. Reflective practice should be encouraged among nurses. It can help nurses process their emotions, identify areas for personal growth, and develop resilience in coping with patient deaths.

6. Healthcare organizations should invest in ongoing professional development opportunities for nurses. This could include workshops, seminars, and training sessions focused on enhancing coping skills, self-awareness, and emotional resilience in the face of patient deaths.

7. Further research should explore the long-term effects of repeated exposure to patient death, how cultural practices and beliefs buffer or intensify nurses' experiences of death related trauma, identify effective supportive strategies, and investigate the unique needs of nurses in different specialties and healthcare settings. This will ensure that future interventions are targeted, context specific, and grounded in evidence.

## Supporting information

**S1 Appendix. Patient death and nurses' coping strategies: Focus group discussion guide.**
(PDF)

**S2 Appendix. Participant information sheet.**
(PDF)

**S3 Appendix. Participant consent form.**
(PDF)

**S4 Appendix. Participant demographic form.**
(PDF)

## Acknowledgments

Kijabe Hospital Nursing leadership is acknowledged for their support in allowing nurses to participate in the study, and all the nurses are appreciated for their willingness to participate and share their experiences.

## Author contributions

**Conceptualization:** Peris Wategi Kiarie, Gabriel Okombo, Dr. Mary B. Adam.

**Data curation:** Peris Wategi Kiarie, Gabriel Okombo, Wambui Makobu, Dr. Joel Ambikile, Dr. Mary B. Adam.

**Formal analysis:** Peris Wategi Kiarie, Gabriel Okombo, Wambui Makobu, Dr. Joel Ambikile, Dr. Mary B. Adam.

**Funding acquisition:** Peris Wategi Kiarie.

**Investigation:** Peris Wategi Kiarie, Gabriel Okombo, Wambui Makobu, Dr. Mary B. Adam.

**Methodology:** Peris Wategi Kiarie, Gabriel Okombo, Dr. Joel Ambikile, Dr. Mary B. Adam.

**Project administration:** Peris Wategi Kiarie.

**Resources:** Peris Wategi Kiarie.

**Software:** Peris Wategi Kiarie.

**Supervision:** Dr. Mary B. Adam.

**Validation:** Peris Wategi Kiarie, Dr. Mary B. Adam.

**Writing – original draft:** Peris Wategi Kiarie.

**Writing – review & editing:** Dr. Mary B. Adam.

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
