## [Decision Letter · Decision Letter 0]

31 Mar 2025

Dear Dr. Kiarie,

Thank you for submitting your manuscript to PLOS ONE. After careful consideration, we feel that it has merit but does not fully meet PLOS ONE’s publication criteria as it currently stands. Therefore, we invite you to submit a revised version of the manuscript that addresses the points raised during the review process.

We look forward to receiving your revised manuscript.

Kind regards,

Michal Soffer

Academic Editor

PLOS ONE

Journal Requirements:

“Kijabe Hospital is a Mission hospital with a focus on underserved populations. An internal small grant of 500 USD was given to cover ethics submissions, and no author received any salary support to do this work.”

4. We note you have included a table to which you do not refer in the text of your manuscript. Please ensure that you refer to Tables 1 to 5 in your text; if accepted, production will need this reference to link the reader to the Table.

Reviewers' comments:

Reviewer's Responses to Questions

**Comments to the Author**

1. Is the manuscript technically sound, and do the data support the conclusions?

Reviewer #1: Yes

Reviewer #2: Partly

Reviewer #3: Partly

2. Has the statistical analysis been performed appropriately and rigorously?

Reviewer #1: N/A

Reviewer #2: N/A

Reviewer #3: N/A

3. Have the authors made all data underlying the findings in their manuscript fully available?

Reviewer #1: Yes

Reviewer #2: Yes

Reviewer #3: Yes

4. Is the manuscript presented in an intelligible fashion and written in standard English?

Reviewer #1: Yes

Reviewer #2: Yes

Reviewer #3: Yes

Reviewer #1: The manuscript titled “Patient death and nurses’ coping strategies” presents a qualitative study investigating the themes of stress and resiliency among nurses in a Kenyan hospital, derived from six focus groups. The methodology and reporting are deemed excellent and offer a comprehensive exploration of the phenomenology of nursing experiences surrounding death. Given the significance of healthcare professional well-being, this reviewer acknowledges the alignment of the study's findings with daily experiences and recognizes the substantial contribution of the detailed reporting to the existing literature. Thus, publication with minor edits is recommended.

Two primary areas warrant further reflection and refinement. Firstly, certain themes and subthemes require differentiation or integration. Specifically, the subtheme “Psychological Trauma” could benefit from differentiation and could be re-labeled as “stress injuries,” employing terminology from the Stress First Aid model for Healthcare Professionals (https://www.ptsd.va.gov/disaster_events/for_providers/stress_first_aid.asp). Precision in terminology is crucial for research. While all encounters with death may technically qualify as trauma, reactions vary, necessitating careful distinction between post-traumatic stress reactions and other responses. The Stress First Aid model identifies four causes: potentially traumatic events, loss/grief, moral injury, and wear and tear, all of which are evident in the submission and carry distinct emotional implications for caregivers. The section on unexpected losses may align with the psychological trauma category, while grief, moral distress, and wear and tear could be presented separately. Furthermore, the “Optimism, pessimism” category appears less substantiated than other themes. Additionally, the sections on religious beliefs and cultural backgrounds could be integrated into a single theme of cultural factors, although this is a minor point. As a Western reader, further elaboration on specific cultural contexts influencing caregiver experiences would enhance the manuscript’s value in informing culturally oriented support.

Secondly, integrating existing literature into the thematic analyses would enhance the manuscript. For instance, the widely researched concept of moral distress/moral injury (Ansari, Warner, Taylor-Swanson, Wilson, Van Epps, Iacob, & Supiano, 2024), which appears in various sections, lacks reference to the substantial body of literature. Similarly, other relevant areas, such as peer support (Scott et al., 2010) and organizational approaches to well-being (Shanafelt & Noseworthy, 2017), are not referenced. Incorporating these concepts into the discussion would enrich the manuscript. This feedback also applies to certain sentences that require more thorough development (i.e. Line 650).

Couple of other comments, line 161 *affirmed needs spelling correction. I prefer APA 7th editor style tables for readability.

Overall, the detailed and extensive coverage of critical nursing experiences represents a significant and positive contribution to scientific knowledge. The depth achieved through the qualitative design is commendable. Therefore, publication with minor revisions is recommended.

Reviewer #2: Thank you for the opportunity to review this manuscript. The focus of the study reported in the paper is to explore the coping mechanisms of nurses in a hospital in Kenya, in relation to their professional experiences with the death of patients. There is potential to the topic, and it is of significance when planning training for professionals. Yet, there are areas which require further attention to provide a more complete argument and report. Hopefully the below will be useful to the authors.

1. The introduction of the paper is rather descriptive and does not provide a more complex layer of how the current knowledge that professionals in health care may struggle with the death of patients surrounds practice. There are many theoretical frames which can support with that. Furthermore, there is no need, in the text, to be comparing degrees of engagement with patients between different professionals in order to emphasise the point that nurses are most engaged and thus affected. This is contested and depends on the context, thus it is important to be examining it in its own merit.

2. Why were nurses who had received palliative care training excluded from the study? What is the reasoning?

3. The methodology needs further attention and the choices made with methods require justification. Why were focus groups the best option for the purposes of this study?

4. Were there interviews as well? This is blurry in the text.

5. The methodology claims saturation of the data, yet how was saturation defined in this study and thus recognised when reached? There were only six focus group discussions. Was it possible to reach saturation?

6. The claim of a grounded theory approach is made. This is not evident in the paper or the methodology.

7. The context was a faith-based hospital, and this needs further unpicking about foundational values of the organisation and how those affect employees in that organisation. Was faith a bigger part in this study?

8. The discussion is not synthesising the findings nor does it offer inferences from the study currently. There is a need to re-work on it, not from the perspective of a summary of the findings.

9. Why are the participant characteristics reiterated in the discussion?

Reviewer #3: Introduction:

The introduction provides relevant literature on the topic. However, the authors acknowledge that the subject has been studied extensively, albeit not sufficiently in the specific context of this study. This point could be more strongly emphasized to justify the study's contribution.

Methods:

• L. 141: The reported number of participants in some of the focus groups is unusually high (14?) based on standard qualitative methodology. A reference supporting this methodological choice is needed, along with an explanation of how such a large focus group was managed.

• What was the duration of each focus group session?

• L. 149: The focus group guide was not provided. Please clarify if this was included in an appendix or supplementary material.

• There is no mention of audio recordings. If recordings were not made, what does "the information was transcribed verbatim" mean?

• Clarify the researchers' backgrounds and their prior relationships, if any, with the participants.

• Grounded theory was mentioned as the guiding framework for data analysis, but further explanation is required:

1. The abstract refers to inductive content analysis rather than grounded theory.

2. The rationale for choosing grounded theory in this study is unclear.

3. Grounded theory involves open, axial, and selective coding. The manuscript does not specify whether these steps were followed in the analysis.

• Excel software was used for data management and analysis—please elaborate on how Excel was utilized for qualitative analysis.

Results:

The findings are interesting and provide useful insights. However, there are several areas for improvement:

• Many paragraphs in the results section lack direct quotes from participants (e.g., L. 182, L. 184—these are just examples).

• There is inconsistency in the use of quotation marks within and outside of participants' direct quotes (e.g., L. 187). A standardized format should be applied.

• Instead of labeling sub-themes as "Sub-theme one," "Sub-theme two," etc., numbering them explicitly would enhance clarity.

• The theme Religious beliefs and spiritual aspects is underdeveloped despite its apparent relevance to the study’s regional context. This should be expanded.

• L. 195: "Sense of denial"—denial of what? Clarification is needed.

• The sub-theme Work environment is comprehensive and contains valuable data on nurses' working conditions. However, some content seems tangential to the study’s main research question (e.g., L. 508—"Nurses discussed the impact of workload on their well-being, citing difficulties in taking breaks"). A clearer link to the study’s focus should be established.

• The study includes nurses from various hospital departments, including the emergency department, medical and surgical wards, intensive care unit, and maternity ward. Is there room to differentiate between the experiences of nurses from different departments? I suggest being more specific. For example, how do the experiences of nurses in high-intensity units like the emergency department and intensive care unit compare to those in medical and surgical wards? Are coping strategies influenced by the nature of the department? Addressing these distinctions could strengthen the discussion and provide deeper insights.

Discussion:

• The discussion is overly long and structured around each sub-theme, rather than focusing on broader conceptual insights. A more synthesized, thematic discussion would enhance readability and impact.

• L. 577-584: belong to the results section (Participants characteristics). In qualitative research, participant characteristics are sometimes included in the methods section. This depends on the journal's policy.

• Some sections of the discussion repeat findings (e.g., L. 666-693, L. 594-599). The discussion should engage with the literature in a way that builds on the findings rather than reiterating them.

• The manuscript frequently cites studies that align with the results (e.g., L. 608, L. 613). However, instead of simply stating that findings are similar, the discussion should explore the implications of these comparisons. Examples:

o L. 609: "These findings align with a study conducted with pediatric nurses in a large women’s and child healthcare hospital in Singapore." How does this comparison deepen our understanding of the current study's findings?

o L. 644: "A review article indicated that there is limited research in nursing examining the role of culture on the impact of death anxiety." How does this relate to Kenya's unique cultural context?

• L. 774-794: This section primarily lists relevant studies rather than engaging in a deeper discussion. The discussion should critically analyze these studies in relation to the findings of the current research.

Study Limitations:

I suggest being more specific. For example,

• The study includes nurses from various hospital departments, including the emergency department, medical and surgical wards, intensive care unit, and maternity ward. I suggest addressing this aspect in the study's limitations as well. Should this be considered a limitation or a strength of the study?

• An additional limitation should be acknowledged: the study was conducted in a single hospital, which may limit the generalizability of the findings.

References:

• The reference list requires careful proofreading. For example, citations 4, 6, and 28 contain formatting errors.

Final Recommendation:

The study presents important findings, but substantial revisions are needed before publication. Specifically, the methodology should be clarified, the results should be better integrated into the discussion, and the manuscript should provide a more conceptually driven interpretation of the findings. I appreciate the opportunity to review this work.

**Do you want your identity to be public for this peer review?** For information about this choice, including consent withdrawal, please see our Privacy Policy

Reviewer #1: **Yes: ** Jake Van Epps

Reviewer #2: No

Reviewer #3: No

---

## [Author Response · Author response to Decision Letter 1]

14 May 2025

Response to Academic Editor

Response: Thank you for the reminder. We have reviewed the PLOS ONE formatting style guides and have ensured that the manuscript now adheres to the journal’s style requirements, including file naming conventions, title formatting, author affiliations, and manuscript structure. The updated files have been renamed and formatted accordingly.

“Kijabe Hospital is a Mission hospital with a focus on underserved populations. An internal small grant of 500 USD was given to cover ethics submissions, and no author received any salary support to do this work.”

Response: Thank you for your guidance. We confirm that the funders had no role in the study. We have included the following statement in our cover letter:

Role of Funder Statement:

Response: Thank you for highlighting this. We have amended the title to ensure consistency between the online submission form and the manuscript.

4. We note you have included a table to which you do not refer in the text of your manuscript. Please ensure that you refer to Tables 1 to 5 in your text; if accepted, production will need this reference to link the reader to the Table.

Response: Thank you for your observation. We have revised the manuscript to ensure that all tables (Tables 1 to 5) are appropriately cited in the text at relevant points, in accordance with PLOS ONE guidelines. (Line 627, 635, 715, 736, & 768).

Reviewer #1

1. Two primary areas warrant further reflection and refinement. Firstly, certain themes and subthemes require differentiation or integration. Specifically, the subtheme “Psychological Trauma” could benefit from differentiation and could be re-labeled as “stress injuries,” employing terminology from the Stress First Aid model for Healthcare Professionals (https://www.ptsd.va.gov/disaster_events/for_providers/stress_first_aid.asp). Precision in terminology is crucial for research. While all encounters with death may technically qualify as trauma, reactions vary, necessitating careful distinction between post-traumatic stress reactions and other responses. The Stress First Aid model identifies four causes: potentially traumatic events, loss/grief, moral injury, and wear and tear, all of which are evident in the submission and carry distinct emotional implications for caregivers. The section on unexpected losses may align with the psychological trauma category, while grief, moral distress, and wear and tear could be presented separately.

Response: Thank you for your insightful feedback regarding the use of precise terminology in describing nurses’ emotional responses to patient death. In response, we have revised the subtheme title from “Psychological Trauma” to “Stress Injuries” to better reflect the spectrum of emotional and psychological responses, aligning our terminology with the Stress First Aid (SFA) model for healthcare professionals. (Line 198, 218, 220, 632 & 638). While we chose not to formally subdivide this theme into distinct subcategories, we have revised the section to explicitly acknowledge the various dimensions of stress outlined in the SFA framework—namely grief/loss, moral distress, wear and tear, and potentially traumatic events. This approach preserves thematic coherence while better capturing the complexity of nurses’ emotional burdens.

Additionally, we have incorporated a brief overview of the SFA model in the literature review section to provide theoretical grounding and support for the use of this terminology. (Line 96).

2. Furthermore, the “Optimism, pessimism” category appears less substantiated than other themes.

i. Additionally, the sections on religious beliefs and cultural backgrounds could be integrated into a single theme of cultural factors, although this is a minor point. As a Western reader, further elaboration on specific cultural contexts influencing caregiver experiences would enhance the manuscript’s value in informing culturally oriented support.

Response: Thank you for this observation. In response, we have strengthened the discussion of the “Optimism and Pessimism” subtheme to ensure clearer synthesis. (Line 664-680).

i. Thank you for the suggestion to combine the “religious beliefs” and “culture” subthemes. We agree these domains are closely linked. However, due to the distinct and significant role that spirituality plays in the study context—as both a belief system and a primary coping mechanism—we have chosen to retain them as separate subthemes. This allows us to honor the weight participants placed on spiritual interpretations of death, beyond cultural customs alone.

ii. Secondly, integrating existing literature into the thematic analyses would enhance the manuscript. For instance, the widely researched concept of moral distress/moral injury (Ansari, Warner, Taylor-Swanson, Wilson, Van Epps, Iacob, & Supiano, 2024), which appears in various sections, lacks reference to the substantial body of literature. Similarly, other relevant areas, such as peer support (Scott et al., 2010) and organizational approaches to well-being (Shanafelt & Noseworthy, 2017), are not referenced. Incorporating these concepts into the discussion would enrich the manuscript. This feedback also applies to certain sentences that require more thorough development (i.e. Line 650).

Response: Thank you for your valuable feedback and for highlighting areas where the integration of existing literature could strengthen the manuscript. We appreciate your suggestions and have made the following revisions:

1. Incorporation of Moral Distress/Moral Injury Literature:

We have integrated the literature on moral distress and moral injury into the thematic analysis, specifically referencing the work by Wilson et al. (2024) (in place of Ansari et al. (2024) as we had difficult accessing the article) to support the discussion of the emotional and ethical challenges faced by nurses. (Line 100). This reference has been included in relevant sections, particularly where we discuss the psychological impact of patient death and the associated moral dilemmas (Line 653).

2. Peer Support and Organizational Approaches to Well-Being:

We have also included references to studies on peer support (Scott et al., 2010) and organizational approaches to well-being (Shanafelt & Noseworthy, 2017). These references enhance the discussion on strategies to mitigate the emotional toll of patient death, including the importance of peer support systems and institutional support that foster nurse well-being. (Line 742, 786).

iii. Couple of other comments, line 161 *affirmed needs spelling correction. I prefer APA 7th editor style tables for readability.

Response: Thank you for your additional comments. In regard to spelling correction, we have reviewed the manuscript for any spelling errors to ensure accuracy. We have also revised the tables labeling to ensure they conform to PLOS ONE formatting guidelines.

Reviewer # 2

1. The introduction of the paper is rather descriptive and does not provide a more complex layer of how the current knowledge that professionals in health care may struggle with the death of patients surrounds practice. There are many theoretical frames which can support with that. Furthermore, there is no need, in the text, to be comparing degrees of engagement with patients between different professionals in order to emphasize the point that nurses are most engaged and thus affected. This is contested and depends on the context; thus, it is important to be examining it in its own merit.

Response: Thank you for your thoughtful feedback on the introduction section of the manuscript. We appreciate your insights and have made the following revisions to address your comments:

i. Complexity of the Current Knowledge: we have revised the introduction to provide a more nuanced understanding of how healthcare professionals, particularly nurses, experience patient death. Instead of simply stating that nurses are most affected, we have highlighted that the emotional burden varies depending on the role and proximity to the patient. (Line 52, 55-59).

ii. Comparing Engagement Between Professionals: In response to your suggestion, we have removed the comparison between different professionals in terms of their degree of engagement with patients. We acknowledge that this is a contested area and that such comparisons may not always be valid. Instead, the focus now remains on nurses' experiences, allowing the discussion to stand on its own merits. (Line 52, 55-59).

iii. Enhancing Theoretical Support: We have incorporated additional references to theoretical frameworks and research on topics such as moral distress, coping mechanisms, and emotional resilience. (Line 935, 951).

iv. Clarity and Flow: We have revised several sentences for clarity, ensuring a smoother transition between ideas, and more explicitly linking factors such as personal attributes, work environment, and training to nurses' emotional responses to death.

v. Training and Coping Mechanisms: We have further emphasized the importance of training in preparing nurses for the emotional challenges of death and dying.

2. Why were nurses who had received palliative care training excluded from the study? What is the reasoning?

Response: Thank you for your thoughtful question regarding the exclusion of nurses with palliative care training from the study.

The exclusion of nurses with palliative care training was intentional to focus on nurses without specialized training, under the assumption that those working in critical care settings would have developed coping mechanisms through their general nursing experience and exposure to patient death. This approach allowed the study to explore the emotional and psychological impact on nurses without formal palliative care training, but who still face similar challenges in critical care environments. The revised explanation clarifies the rationale for this decision and the study's focus on a broader perspective. (Line 149-154).

3. The methodology needs further attention and the choices made with methods require justification. Why were focus groups the best option for the purposes of this study?

Response:

Thank you for this insightful comment. We have revised the "Design, Sample and Data Collection" section to include a brief rationale for our methodological choices. Specifically, we justified the use of a qualitative phenomenological design to explore the deeply personal and emotional experiences of nurses coping with patient death. Additionally, we have clarified that focus group discussions (FGDs) were selected as the primary method of data collection because they promote open discussion, allow participants to reflect on shared experiences, and generate rich data through group dynamics. The revised section also notes that while traditional FGDs are smaller, larger groups were managed effectively with careful moderation to ensure quality and depth of data collection. (Line 161-162).

4. Were there interviews as well? This is blurry in the text.

Response: Thank you for pointing this out. We confirm that the study used only focus group discussions (FGDs) for data collection; no individual interviews were conducted. To eliminate ambiguity, we have revised the methods section to explicitly state that FGDs were the sole method of data collection and that no one-on-one interviews were used in this study. This information has been added on the manuscript. (Line 159).

5. The methodology claims saturation of the data, yet how was saturation defined in this study and thus recognized when reached? There were only six focus group discussions. Was it possible to reach saturation?

Response: Thank you for this important observation. Saturation in this study was defined as the point at which no new themes, categories, or insights were emerging from subsequent focus group discussions. After the fifth FGD, the research team began to observe significant repetition of patterns, ideas, and experiences across participants. The sixth FGD confirmed thematic redundancy, affirming that the core issues had been adequately captured. Therefore, the decision to conclude data collection at six FGDs was based on this observed saturation, aligning with qualitative research principles. This information has been added on the manuscript. (Line 176-178).

6. The claim of a grounded theory approach is made. This is not evident in the paper or the methodology.

Response: We appreciate the editor’s observation. Upon review, we acknowledge that the claim of using a grounded theory approach was inaccurate and not aligned with the study’s actual methodology. The study followed a phenomenological design to explore and understand the lived experiences of nurses in coping with patient death. To analyze the qualitative data, we used inductive content analysis- an approach suitable for identifying themes in phenomenological studies where researchers derive categories from raw data without pre-imposed coding frameworks. The text has been revised to ensure consistency and remove any incorrect references to grounded theory. (Line 139-142).

7. The context was a faith-based hospital, and this needs further unpicking about foundational values of the organization and how those affect employees in that organization. Was faith a bigger part in this study?

Response: Thank you for this insightful comment. The context of the faith-based hospital indeed shaped the working environment and may have influenced how nurses experienced and coped with patient death. While the study did not set out to explore the influence of faith explicitly, we recognize that spiritual values may have informed participants' coping mechanisms and their perceptions of care and loss. To address this, we have included a brief contextual explanation of the institution's faith-based foundation and reflected on how this may have influenced participants' experiences in the discussion section. (Line 128-129;690-695).

8. The discussion is not synthesizing the findings nor does it offer inferences from the study currently. There is a need to re-work on it, not from the perspective of a summary of the findings.

Response: Thank you for the insightful comment. We acknowledge that the initial discussion section leaned toward summarizing the findings rather than synthesizing them or offering deeper inferences. In response, we have revised the discussion to more clearly interpret the findings in light of existing literature, highlight emerging patterns, and offer theoretical and practical implications for nursing practice, particularly in resource-limited, faith-based healthcare settings. The revised discussion now moves beyond summary to critically explore what the findings mean, how they connect with broader issues in end-of-life care, and what they suggest for future improvements. (Line 637-809).

9. Why are the participant characteristics reiterated in the discussion?

Response: Thank you for your valuable feedback regarding the placement of participant characteristics. We have revised the ma

---

## [Decision Letter · Decision Letter 1]

16 Jun 2025

PONE-D-24-53361R1

Patient Death and Nurses’ Coping Strategies: Perception of Nurses at a Tertiary Referral Hospital in Kenya

PLOS ONE

Dear Dr. Kiarie,

Thank you for submitting your manuscript to PLOS ONE. After careful consideration, we feel that it has merit but does not fully meet PLOS ONE’s publication criteria as it currently stands. Therefore, we invite you to submit a revised version of the manuscript that addresses the points raised during the review process.

Please submit your revised manuscript by Jul 31 2025 11:59PM.  If you will need more time than this to complete your revisions, please reply to this message or contact the journal office at plosone@plos.org . Please include the following items when submitting your revised manuscript:

We look forward to receiving your revised manuscript.

Kind regards,

Michal Soffer

Academic Editor

PLOS ONE

Journal Requirements:

Reviewers' comments:

Reviewer's Responses to Questions

**Comments to the Author**

1. If the authors have adequately addressed your comments raised in a previous round of review and you feel that this manuscript is now acceptable for publication, you may indicate that here to bypass the “Comments to the Author” section, enter your conflict of interest statement in the “Confidential to Editor” section, and submit your "Accept" recommendation.

Reviewer #2: (No Response)

Reviewer #3: (No Response)

2. Is the manuscript technically sound, and do the data support the conclusions?

Reviewer #2: Yes

Reviewer #3: Yes

3. Has the statistical analysis been performed appropriately and rigorously?

Reviewer #2: N/A

Reviewer #3: N/A

4. Have the authors made all data underlying the findings in their manuscript fully available?

Reviewer #2: Yes

Reviewer #3: No

5. Is the manuscript presented in an intelligible fashion and written in standard English?

Reviewer #2: Yes

Reviewer #3: Yes

6. Review Comments to the Author

Reviewer #2: (No Response)

Reviewer #3: (No Response)

7. PLOS authors have the option to publish the peer review history of their article (what does this mean? ). If published, this will include your full peer review and any attached files.

**Do you want your identity to be public for this peer review?**  For information about this choice, including consent withdrawal, please see our Privacy Policy .

Reviewer #2: No

Reviewer #3: No

---

## [Author Response · Author response to Decision Letter 2]

28 Jul 2025

Reviewer’s comments

Introduction:

105 – 116:

This important paragraph needs references. For example:

"… nurses working in low- and middle-income countries (LMICs) like Kenya, often operate under resource constrained environments, with limited access to psychological support, continuing professional 108 development, or structured debriefing sessions following patient deaths" (ref).

"Kenya presents a unique context due to its cultural diversity and varying beliefs surrounding death…" (ref)

Response: Thank you for this observation. We have reviewed the reference list to ensure accuracy and completeness. In response to the comment on the following sentence:

“…nurses working in low- and middle-income countries (LMICs) like Kenya, often operate under resource-constrained environments, with limited access to psychological support, continuing professional development, or structured debriefing sessions following patient deaths…”

We have now supported this statement with a relevant reference: Afulani, P. A., Ongeri, L., Kinyua, J., Temmerman, M., Mendes, W. B., & Weiss, S. J. (Year). Psychological and physiological stress and burnout among maternity providers in a rural county in Kenya: Individual and situational predictors. Reference No. 25. (Line: 933-935).

In response to the comment on the following sentence:

"Kenya presents a unique context due to its cultural diversity and varying beliefs surrounding death…" (ref)

We have now supported this statement with a relevant reference: Gire, J. (2014). How death imitates life: Cultural influences on conceptions of death and dying. Online Readings in Psychology and Culture, 6(2). Reference list No. 26. (Line: 936-938).

Methodology:

In my opinion, there appears to be a confusion between the content analysis approach—which typically does not involve the development of themes—and thematic analysis. The reference cited by the authors clearly explains the distinctions between these two methods. I also question the appropriateness of using content analysis within a phenomenological framework.

Response: Thank you for the valuable observation. We acknowledge the conceptual distinctions between content analysis and thematic analysis, especially regarding their philosophical underpinnings and analytic procedures. After revisiting both our methodological approach and the cited reference (Vears & Gillam, 2022), we recognize that our original phrasing may have created confusion.

Our analysis followed an inductive coding process that led to the generation of themes, consistent with thematic analysis. Therefore, we have revised the abstract and methodology sections to clearly reflect that thematic analysis using inductive coding was employed. (Line: 30-31; 194-201).

Discussion:

1. In the revised version, each theme is still followed by a summary of the findings. In my opinion, it is unnecessary, as it repeats content that has already been presented in the findings. For example, in line 630 – 635: "The theme of individual process provided a comprehensive understanding of the individual factors that characterizes/influence nurses' reactions, and emotional responses to patient death. This theme is outlined through the following sub themes: Stress injuries, Optimism and Pessimism, Religious Belief, Culture and Self-management. It also highlights the individualized and self-directed approaches nurses adopt to navigate the emotional complexities associated with their patient deaths, particularly in choosing coping strategies “Table 2”. "

Response: Thank you for the helpful observation. We have carefully revised the discussion section and removed summaries of the findings that were previously restated under each theme. (Line: 630-631; 708;727;756-757).

2. The discussion is thorough and detailed; however, it is still organized according to themes and sub-sub-themes. Perhaps this is a matter of writing style. In any case, I believe it would have been more effective to identify a few key points and develop a deeper discussion around them, rather than addressing each finding separately.

Response: Thank you for your observation. We have integrated some of the subthemes that have related points and discussed them together. (Line: 759; 787).

Conclusion: The content is repetitive. Paragraph 1 (lines 811–818) repeats findings that have already been discussed. Paragraph 2 content is detailed in the following implications section.

• Response: We sincerely appreciate the reviewers’ thoughtful feedback regarding the organization and content of our conclusion section. We have carefully revised this section to address the concerns about repetition. We have removed restatements of findings that were already thoroughly discussed in the results and discussion sections. (Line: 797-807)

---

## [Editor Report · Decision Letter 2]

7 Aug 2025

Dear Dr. Kiarie,

We look forward to receiving your revised manuscript.

Kind regards,

Lily Kpobi, Ph.D.

Academic Editor

PLOS ONE

Journal Requirements:

2, Please review your reference list to ensure that it is complete and correct. If you have cited papers that have been retracted, please include the rationale for doing so in the manuscript text, or remove these references and replace them with relevant current references. Any changes to the reference list should be mentioned in the rebuttal letter that accompanies your revised manuscript. If you need to cite a retracted article, indicate the article’s retracted status in the References list and also include a citation and full reference for the retraction notice.

**Section Editor's Comments:**

Please effect the changes to the discussion as suggested by reviewer 3 at revision 1. The discussion section must not be sub-divided by themes and sub-themes, but must be written as a single narrative which discusses the findings and their implications.
---

## [Author Response · Author response to Decision Letter 3]

29 Sep 2025

Reviewer’s comments

Please effect the changes to the discussion as suggested by reviewer 3 at revision 1.

The discussion section must not be sub-divided by themes and sub-themes, but must be written as a single narrative which discusses the findings and their implications.

Response: Thank you for this observation. The Discussion section has been restructured into a single, cohesive narrative. The word count has been reduced from 2,029 to 1,895 words (lines 626–779).

---

## [Decision Letter · Decision Letter 3]

10 Dec 2025

Patient Death and Nurses’ Coping Strategies: Perception of Nurses at a Tertiary Referral Hospital in Kenya

PONE-D-24-53361R3

Dear Dr. Kiarie,

We’re pleased to inform you that your manuscript has been judged scientifically suitable for publication and will be formally accepted for publication once it meets all outstanding technical requirements.

Kind regards,

Vinit Kumar Ramawat, M.Sc Nursing

Guest Editor

PLOS One

Additional Editor Comments (optional):

Reviewers' comments:

Reviewer's Responses to Questions

**Comments to the Author**

Reviewer #4: All comments have been addressed

2. Is the manuscript technically sound, and do the data support the conclusions?

Reviewer #4: Yes

3. Has the statistical analysis been performed appropriately and rigorously?

Reviewer #4: N/A

4. Have the authors made all data underlying the findings in their manuscript fully available?

Reviewer #4: Yes

5. Is the manuscript presented in an intelligible fashion and written in standard English?

Reviewer #4: Yes

Reviewer #4: Overall, a good study has been conducted and an important issue has been examined, and its publication would be useful.

**Do you want your identity to be public for this peer review?** For information about this choice, including consent withdrawal, please see our Privacy Policy

Reviewer #4: **Yes: ** Alireza Nikbakht Nasrabadi

---

## [Editor Report · Acceptance letter]

PONE-D-24-53361R3

PLOS One

Dear Dr. Kiarie,

I'm pleased to inform you that your manuscript has been deemed suitable for publication in PLOS One. Congratulations! Your manuscript is now being handed over to our production team.

Kind regards,

on behalf of

Prof. Vinit Kumar Ramawat

Guest Editor

PLOS One